# Enhanced Photoluminescence and Electrical Properties of n-Al-Doped ZnO Nanorods/p-B-Doped Diamond Heterojunction

**DOI:** 10.3390/ijms23073831

**Published:** 2022-03-30

**Authors:** Yu Yao, Dandan Sang, Liangrui Zou, Dong Zhang, Qingru Wang, Xueting Wang, Liying Wang, Jie Yin, Jianchao Fan, Qinglin Wang

**Affiliations:** 1Shandong Key Laboratory of Optical Communication Science and Technology, School of Physics Science and Information Technology, Liaocheng University, Liaocheng 252000, China; lcuyaoyu0814@163.com (Y.Y.); zouliangruilcu@163.com (L.Z.); zhangdong@lcu.edu.cn (D.Z.); wangqingru@lcu.edu.cn (Q.W.); wangxueting@lcu.edu.cn (X.W.); 2Key Laboratory of Advanced Structural Materials Ministry of Education, School of Materials Science and Engineering, Changchun University of Technology, Changchun 130012, China; wangliying@ccut.edu.cn; 3School of Material Science and Engineering, Liaocheng University, Liaocheng 252000, China; yinjieily@163.com; 4Shandong Liaocheng Laixin Powder Materials Science and Technology Co., Ltd., Liaocheng 252000, China; 13676388216@126.com

**Keywords:** aluminum-doped ZnO NRs, B-doped diamond, heterojunction, photoluminescence, electrical transport behavior

## Abstract

The hydrothermal approach has been used to fabricate a heterojunction of n-aluminum-doped ZnO nanorods/p-B-doped diamond (n-Al:ZnO NRs/p-BDD). It exhibits a significant increase in photoluminescence (PL) intensity and a blue shift of the UV emission peak when compared to the n-ZnO NRs/p-BDD heterojunction. The current voltage (*I-V*) characteristics exhibit excellent rectifying behavior with a high rectification ratio of 838 at 5 V. The n-Al:ZnO NRs/p-BDD heterojunction shows a minimum turn-on voltage (0.27 V) and reverse leakage current (0.077 μA). The forward current of the n-Al:ZnO NRs/p-BDD heterojunction is more than 1300 times than that of the n-ZnO NRs/p-BDD heterojunction at 5 V. The ideality factor and the barrier height of the Al-doped device were found to decrease. The electrical transport behavior and carrier injection process of the n-Al:ZnO NRs/p-BDD heterojunction were analyzed through the equilibrium energy band diagrams and semiconductor theoretical models.

## 1. Introduction

Because of its extensive uses in blue, violet and ultraviolet optoelectronic and field emission devices, wide bandgap semiconductors have attracted a lot of attention in recent years [1,2]. ZnO, which has a large bandgap (3.37 eV) and significant activation energy (60 meV) at ambient temperature, is regarded as a viable candidate material in the optoelectronic application sector [3] due to its lower laser threshold and increased optical efficiency. Because nanostructured ZnO (nanorods, nanowires, nanobelts and nanosheets) offers higher optoelectronic capabilities than bulk materials, it has attracted a lot of attention [4,5,6,7]. However, establishing homojunction based on ZnO remains a significant problem due to the difficulty in synthesizing a stable p-type of ZnO [8]. Researchers are attempting to develop a ZnO-based heterojunction as a remedy for this problem. Some p-type substrate materials have recently been employed in the fabrication of ZnO-based heterojunctions, including SiC [8,9], Si [10,11], GaN [12,13], CuO [14], graphene [15], boron-doped diamond (BDD) [16,17,18,19] and NiO [20]. Compared with conventional materials such as Si and GaN, BDD exhibits excellent properties such as high thermal conductivity, wide forbidden band (5.47 eV), high breakdown voltage (~10^7^ V/cm), high electron hole mobility (2400 cm^2^/(V·s)), chemical stability and radiation resistance at room temperature, and their thermal stability is compatible for the construction of functional components or devices which require high (above 400 °C) processing [21]. As a result, BDD-based devices are durable and reliable in harsh environments such as high temperature, high frequency and high power. Doping diamond with boron leads to a semiconductor or metallic states; a large density of boron impurities must be incorporated into the lattice in order to achieve sufficient conductivity [22]. The typical crystal structures of BDD is shown in Figure 1a. However, there are only a few reports in the literature on the optical and electrical properties of n-ZnO/p-BDD heterojunction devices, such as high-temperature negative resistance behavior [23], high-temperature electrical transport behavior [18,24], UV photoelectric capabilities [16] and photocatalytic activity [25,26]. However, no reports on the optical and electrical properties of intentionally doped n-type ZnO nanostructures on the p-type diamond have been published so far. Due to changes in its composition, microstructure and defects, doped ZnO exhibits unique optical and electrical properties such as low resistivity, high visible light transmittance, and high carrier concentration. As a result, the optical and electrical properties of ZnO-based heterojunctions can be further modified. From this perspective, doping appears to be a simple and effective technique to alter the optical and electrical properties of the n-ZnO/p-BDD heterojunction.

Group III elements are often used as dopants in ZnO, with Al being one of the most commonly used dopants due to its small ionic radius and inexpensive cost [27]. Al-doped ZnO are replaced with aluminum atoms and are situated in the tetrahedral gaps generated by four nearby oxygen atoms (Figure 1b) and have more valence electrons than pure ZnO, which means they have more loosely bound electrons that can quickly break away and become conduction band electrons. As a result, Al-doped ZnO carrier concentration will be high, impacting both its optical and electrical properties. In this research, we examine the photoluminescence (PL) and electrical properties of an n-Al:ZnO NRs/p-BDD heterojunction. The physical mechanism of heterojunction carrier transport behavior is investigated using energy band diagrams and semiconductor theoretical models, suggesting the theoretical framework for the development of novel high-temperature resistant white light-emitting diodes (WLEDs).

## 2. Results and Discussion

Figure 2a–c show the SEM images of BDD films, ZnO NRs and Al:ZnO NRs deposited on BDD substrate, respectively. The ZnO NRs and Al:ZnO NRs were grown almost vertically on the diamond substrate (confirmed by the image in Figure 2b,c), and the average diameters (lengths) of these rods are 500 nm (2 μm) and 300 nm (1.2 μm), respectively. From the side, it reflects that the ZnO NRs become smaller when doped with Al. BDD films are made up of small diamond grains in pyramid forms with sizes ranging from 1 to 3 μm. The Al:ZnO NRs show the typical hexagonal fibrous zincite structure. To investigate the elemental distribution of Al:ZnO NRs/BDD heterojunction, the SEM images were analyzed as shown in Figure 2d. The EDS mapping images of Zn, O, Al, C and B elements correspond to Figure 2d1–d5, respectively. It is observed that the main elements on the Al:ZnO NRs are Zn and O. The Al elements are highly dispersed on the ZnO NRs, and the Zn and C elements form complementary dispersions. B has a C-element-dependent distribution, which corresponds to the top and bottom structures of Al:ZnO NRs and BDD substrates, respectively. Figure 2e shows the EDS spectra of Al:ZnO NRs/BDD and the results show that the main composition of the heterojunction is composed of Zn, O and C elements with trace amounts of Al and B elements. The elemental composition indicates that Al is homogeneously doped into the ZnO NRs.

The XRD patterns of n-ZnO NRs/p-BDD heterojunction and n-Al:ZnO NRs/p-BDD heterojunction deposited on Si substrates are shown in Figure 3. For the n-ZnO NRs/p-BDD heterojunction, the main peaks located at 32°, 34.5°, 36.4°, 47.7° and 63° belong to the (100), (002), (101), (002) and (103) diffraction peaks of ZnO in that order (JCPDS card no. 36-1451). The remaining diffraction peaks (111), (220) located at 44°, 75.4° are mainly associated with the diamond. The diffraction peak located at 69.32° is calibrated by the Si substrate. When doped with Al, the diffraction peaks at 56.3°, 65.9° are added, and they are designated as (422) (ZnAl_2_O_4_ (JCPDS card no. 05-0669)) and (220) (Al (JCPDS card no. 04-0787)) crystal orientations. These results indicate that Al:ZnO NRs still maintain the structure of hexagonal wurtzite. In addition, some of the crystal orientations associated with ZnO exhibit a decrease in peak intensity, which is likely due to the deterioration of the crystallinity of ZnO nanorods by Al doping. Moreover, the (100), (002) and (103) diffraction peaks are shifted toward lower angles, which may be due to the substitution of Al^3+^ for the Zn^2+^ position in the crystal [28].

Figure 4a shows the XPS scan spectrum (0–1100 eV) of the n-Al:ZnO NRs/p-BDD heterojunction calibrated with the C (1s) peak as the baseline. The peaks of Zn2p, O1s and C1s of the major elements are visible in the spectrum, and the oscillatory peaks of Zn LMM and O KLL are also observed, which indicates the presence of Zn, O and C in the heterojunction. The presence of the C1s peaks in the spectra is due to the diamond substrate in the heterojunction and the contamination caused by the exposure of the sample to the ambient [29]. 

High-resolution XPS spectra of important elements such as Zn, O and Al were performed to better understand the surface chemistry of the n-Al:ZnO NRs/p-BDD heterojunction and the scanned spectra are displayed in the inset of Figure 4. The illustration of Figure 4a shows the high-resolution XPS scan spectrum of Zn2p. The Zn2p spectrum consists of a Zn2p_3/2_ peak centered at a binding energy of 1021.5 ± 0.1 eV and a Zn2p_1/2_ peak centered at a binding energy of 1044.5 ± 0.1 eV, and the difference in binding energy between the two peaks is 23 ± 0.1 eV [30]. In addition, the presence of Zn ions in ZnO can be concluded as Zn^2+^ based on these binding energy values [31]. Figure 4b shows the high-resolution XPS scan spectrum of O1s, where the peak of the O1s peak appears at 531.6 ± 0.1 eV before Gaussian fitting. The O1s peak can be decomposed into three components, centered at 530.4 ± 0.1 eV, 531.6 ± 0.1 eV and 536.1 ± 0.1 eV, and are defined by Gaussian fitting O_I_, O_II_ and O_III_. The lower binding energy O_I_ peak (530.4 eV) can be ascribed to the O^2−^ ion on the wurtzite ZnO structure, which is surrounded by Zn atoms and forms a fully complementary state with the surrounding adjacent O^2−^ ions [32,33]. The O_II_ peak at intermediate binding energy (531.6 eV) is correlated with the O^2−^ ions in the oxygen-deficient region of the ZnO matrix [33]. The intensity of the component further reflects the concentration of oxygen vacancies, which can be assigned to the Zn-O bond and adsorbed oxygen including physically adsorbed or chemisorbed oxygen and hydroxides within the surface layer, respectively [34,35,36]. The highest binding energy O_III_ peak (536.1 eV) is due to the loosely bound oxygen on the surface of the heterojunction or due to adsorbed water and oxygen at the crystal boundaries. Figure 4c shows a binding energy value of 74.1 ± 0.1 eV for Al2p, and after performing a Gaussian fit the main component centered at 73.8 ± 0.1 eV is amorphous Al_2_O_3_, which is much lower than the normal binding energy of 75.6 eV, and may be due to the variation of Al-surrounded atoms within the different lattices; this confirms that Al^3+^ is doped into a ZnO matrix and acts as a donor [34]. It also could be assumed that the Al_2_O_3_ may be segregated to the grain boundaries, leading to the grain boundary barrier [37]. A full-range XPS spectrum of the substrate BDD was performed with a high-resolution scan of the major elements B and C (Figure 4d–f). This spectrum shows a C1s peak at 283.9 eV, a B1s peak at 186.6 eV and an O1s peak at 532 eV, where the C1s peak is mainly determined by graphite-like carbon atoms with *Sp*^2^ hybridization in the C-C bond. The B1s peak within 182–190 eV is mainly associated with the B-C bond and the oxide BC_2_O [38], and the O1s peak at 532 eV mostly originates from surface oxygen contamination or C-OH groups [39].

Undoped ZnO generally contains a variety of intrinsic defects, such as interstitial zinc, zinc vacancies, oxygen vacancies, interstitial oxygen and anti-situated oxygen [40]. In the bandgap, these intrinsic defects will form acceptor or donor energy levels, which will have a significant impact on the luminous performance of ZnO. By introducing the dopant aluminum, the defect environment will change whether aluminum atoms replace zinc atoms or occupy interstitial sites. To further investigate the effect of aluminum doping on n-ZnO/p-BDD heterojunction, the room-temperature PL spectra of n-Al:ZnO NRs/p-BDD heterojunction and n-ZnO NRs/p-BDD heterojunction were measured separately.

Figure 5a shows the PL spectra of n-ZnO NRs/p-BDD heterojunction and n-Al:ZnO NRs/p-BDD heterojunction under the laser excitation at 346 nm. It is observed that both the n-ZnO NRs/p-BDD and the n-Al:ZnO NRs/p-BDD heterojunctions show strong UV luminescence with their luminescence peaks located at 390 nm and 382 nm, respectively. In general, the UV emission peak of ZnO is caused by near-band edge emission, which is attributed to the free exciton complex of ZnO. In the visible region, the two samples show various defect-related emissions centered at ~407 nm, ~430 nm, ~456 nm, ~537 nm, ~565 nm and ~584 nm. Among them, the emission at 407 nm, 430 nm and 456 nm is associated with diamond, and the narrow band peaks at 430 nm and 456 nm are usually observed in natural Ia diamond, and the appearance of these peak sites is attributed to the N3 center [41]. While the green luminescence peaks at 537 nm and 565 nm are mostly due to defects in the ZnO nanostructure, they have also been controversial, and many different explanations for the green luminescence of ZnO have been proposed, including interstitial zinc (Zni), surface defects, single ion oxygen vacancies (Vo+) and double ion oxygen vacancies (Vo++), among others [42]. The most widely held belief is that the green emission of ZnO is caused by the combining of singly ionized oxygen vacancies and photogenerated holes in the ZnO lattice [43]. Compared with the n-ZnO NRs/p-BDD heterojunction, the PL intensity of the n-Al:ZnO NRs/p-BDD heterojunction is significantly enhanced and the UV-emission peak is significantly blue-shifted. When aluminum is doped into ZnO, the band gap of ZnO widens, resulting in the Burstein–Moss blue shift. The combined action of surface plasmon resonance and metal-induced crystallization of the ZnO matrix may be responsible for the increased PL intensity [44]. 

The transformed CIE chromaticity diagram is shown in Figure 5b. The chromaticity coordinates of the n-ZnO NRs/p-BDD heterojunction shift in the white light region with aluminum doping, from (0.2943, 0.3052) to (0.2957, 0.2907), where the emission intensity of the n-Al:ZnO NRs/p-BDD heterojunction is enhanced and the chromaticity coordinates (0.2957, 0.2907) are closer to the standard white light, as shown in the figure. The outstanding broadband WLED emission of the n-Al:ZnO NRs/p-BDD heterojunction makes it an interesting candidate material for WLED devices [45].

The schematic diagram of the n-Al:ZnO NRs/p-BDD heterojunction devices are shown in the top right illustration in Figure 6. Two silver conductors are attached as positive and negative electrodes on the conductive surface of the BDD film and ITO glass, respectively. The *I-V* characteristics of Ag/ITO/Ag and Ag/BDD/Ag show a linear relationship (the bottom left illustration of Figure 6), which indicates ohmic contact. Since the power functions of the two structures are almost the same, Al:ZnO NRs-ITO shows ohmic contact characteristics [46]. From Hall measurements, the carrier concentration (3.8 × 10^17^ cm^−3^), resistivity (0.33 Ω cm) and mobility (48.9 cm^2^ V^−1^ s^−1^) of the BDD were extracted.

The *I-V* characteristic curve of n-Al:ZnO NRs/p-BDD heterojunction and n-ZnO NRs/p-BDD heterojunction is shown in Figure 6. The heterojunction has good rectification characteristics with a high rectification ratio of 838 at 5 V, which is more than 40 times higher than that of the n-ZnO NRs/p-BDD heterojunction (19.3). Such high values indicate that Al doping significantly improves the quality of the device as well as the quality of the crystal interface at the p-n junction. The forward turn-on voltage of n-Al:ZnO NRs/p-BDD heterojunction is only 0.27 V, which is much smaller than n-ZnO NRs/p-BDD heterojunction value (3.4 V). A low turn-on voltage value suggests a better electrical performance of n-Al:ZnO NRs/p-BDD heterojunction which is due to the generation of more free electrons. Under the influence of applied voltage, these free electrons can flow more easily from the conduction band of n-Al:ZnO NRs to the conduction band of BDD, resulting in a drop in the turn-on voltage and an increase in the forward current [47]. The reduction in turn-on voltage and rapid increase in forward current in n-Al:ZnO NRs/p-BDD rectifier devices helps to reduce power losses and improve switching characteristics. In addition, under the reverse bias of −5 V, the reverse leakage current of n-Al:ZnO NRs/p-BDD heterojunction is only 0.077 μA. The decreased leakage current could be due to the NR array’s compactness and/or the bigger nanorod size, which has fewer grain boundaries. It is also likely due to Al doping, which alters the interface state between the ZnO NRs and the BDD and hence suppresses the reverse bias current. It is emphasized that the forward current of the n-Al:ZnO NRs/p-BDD heterojunction shows a sharp increase. When the applied voltage reaches 5 V, the forward current is as high as 67.5 mA, which is more than 1300 times higher than that of the n-ZnO NRs/p-BDD heterojunction. In addition, the turn-on voltage of the n-Al:ZnO NRs/p-BDD device is 3.7 times and 2.4 times lower than that of the reported n-Al:ZnO/p-Si heterojunction [48] and n-Al:ZnO/p-MoSe_2_ heterojunction [49], respectively. In comparison to previously reported ZnO-related heterojunctions (n-ZnO/p-Si [50], n-Ga:ZnO/p-Si [51]), the fabricated n-Al:ZnO NRs/p-BDD heterojunction has improved electrical properties such as lower turn-on voltage, smaller leakage current, greater forward current and rectification ratio. The pn junction parameters of the n-Al:ZnO NRs/p-BDD heterojunction demonstrate overall better *I-V* characteristics with comparatively high performance when compared to other p-type substrates, indicating the good electrical qualities of p-BDD-based heterojunction.

To gain better insight into the electrical behavior of the n-Al:ZnO NRs/p-BDD heterojunction, we propose a schematic diagram of the heterojunction energy band as shown in Figure 7. The barrier height at the conduction band is substantially higher than that at the valence band when n-type ZnO NRs and p-BDD are coupled as a heterojunction, preventing the electron leap at the conduction band side. Therefore, the conductivity of the n-ZnO/p-BDD heterojunction is mainly determined by the holes in the valence band [52]. After doping with Al, the Al:ZnO NRs exhibit more defect levels, mainly including impurity energy levels. The oxygen vacancies on the ZnO surface, as well as the interstitial zinc, account for the majority of the defects. The impurity energy level on the Al:ZnO NR side increases as the forward current increases, and the barrier height falls. Furthermore, the holes on the diamond side can penetrate the impurity energy level via the tunneling effect, resulting in a higher tunneling current. Furthermore, Al doping enhances the surface defect concentration of ZnO, and the high density of oxygen defects traps the holes created by excitation, promoting electron-hole pair diffusion and separation, hindering recombination and increasing the lifespan of emitted carriers. The free electrons that are unable to recombine with the hole pair are injected from the ZnO conduction band into the diamond conduction band, thus increasing free carrier concentration. Moreover, after the Al is doped in the n-Al:ZnO NRs/p-BDD heterojunction, the bandgap of Al:ZnO NRs is extended to lower the heterojunction’s barrier height and makes carrier injection easier. As a result, the n-Al:ZnO NRs/p-BDD heterojunction shows better electrical properties that include a large forward current and a low turn-on voltage, which improves carrier injection efficiency.

The semi-logarithmic plot of the *I-V* characteristic curve of the n-Al:ZnO NRs/p-BDD heterojunction is shown in Figure 8a. According to the diode ideal Equation (1):(1)$$I=I_{s}\left[exp\left(\frac{qV}{nkT}-1\right)\right]$$
where *n* is the ideal coefficient, i.e., the linear part of the *I-V* characteristic, *I_s_* is the reverse saturation current, *q* is the electronic charge, *V* is the applied voltage, *T* is the absolute temperature and *k* is the Boltzmann constant. For the specified voltage region of 0–0.5 V, in the range corresponding to the device turn-on voltage for fitting, the ideality factor, *n* for n-Al:ZnO NRs/p-BDD was obtained as 6.8. An n value greater than 2 indicates the deviation from the ideality factor for p-n heterojunction [53]. This may arise from the presence of a large number of surface states in the heterojunction, space charge-limited conduction inside the device, deep-level assisted tunneling or parasitic rectification junctions [54]. However, although the *n*-value is not optimal, it is significantly lower than the reported n-ZnO NRs/p-BDD heterojunction (8.6) [24]. This can be attributed to Al-doped ZnO generating more carriers in the heterojunction, changing the depletion layer of the heterojunction and leading to more attempted penetration effects and carrier generation–combination processes.

The barrier height (*Φ**_B_*) is estimated from Equation (2):(2)$$I_{s}=AA^{*}\cdot exp\left(\frac{-q\Phi_{B}}{kT}\right)$$
where *A* is the contact area, *A** is the effective Richardson constant (32 A/cm^2^ K^2^) based on ZnO. Bringing the value of the reverse saturation current *I_S_* obtained in Equation (1) into Equation (2), we can obtain the values of barrier heights of 0.59 eV and 0.55 eV for n-ZnO NRs/p-BDD heterojunction and n-Al:ZnO NRs/p-BDD heterojunction, respectively. The latter exhibits a lower barrier height than the former, consistent with the energy band analysis above that the barrier height is lower concerning the n-Al:ZnO NRs/p-BDD heterojunction. The reduction of the barrier height is mainly due to doping and structural defects. As a result of the metal-induced bandgap state determining the potential discontinuity at the defined junction, the Fermi energy level is mainly determined by the intrinsic interface properties rather than by the constituent materials forming the interface [55].

To investigate the injection mechanism of tunneling current in depth, the *I-V* characteristics at room temperature are analyzed using the Fowler–Nordheim (F-N) model (Figure 8b). Charge injection usually takes place through interfacial barrier tunneling when there is insufficient thermal energy in the low bias region to help the electrons in overcoming the barrier height at room temperature. As a result, the direct tunneling effect is commonly assumed to occur. As the bias voltage increases, the barrier height decreases, when the conduction mechanism changes to FN tunneling [56]. The FN current is given by the following FN Equation (3) [57]:(3)I∝V2exp[−4d(qϕ)32(2m*)123qℏV]
where *ϕ* is the energy potential barrier height, *d* is the distance required for tunneling shuttle, ℏ is Planck’s constant divided by 2π and *m** is the effective mass of the charge carrier. Figure 8b shows a plot of the relationship between ln(*I/V*^2^) and *1/V*. The introduction of two alternative transmission modes, which appear sequentially before and after the voltage inflection point (*V*_t_), is the plot’s most notable characteristic. The *V*_t_ of this heterojunction is found to be 4 V. When *V*_t_ > 4.3 V, it is observed that the linear relationship of the curves in the figure shows a negative slope, indicating the F-N tunneling effect. When *V*_t_ < 4.3 V, the electrical transport properties change and the curves become logarithmic with 1/*V*, which indicates a direct tunneling effect.

Figure 8c shows the logarithmic *I-V* characteristic curves of the n-Al:ZnO NRs/p-BDD heterojunction. The plot can be divided into three regions based on the applied bias voltage. At low forward voltage (region I, 0 < *V* < 1.7 V), the fit yields *I~V*^1.4^, an exponential value close to 1, indicating that the *I-V* characteristic curve at this point follows the linear relationship of Ohm’s law. In moderate forward voltage (region II, 1.7 V < *V* < 5.8 V), the forward current at this time generally follows the I-exp(*αV*) relationship due to the recombination-tunneling mechanism of wide-bandgap semiconductor heterojunction devices. The injection efficiency constant α was calculated by fitting to be 0.5 V^−1^, which lies in the range of the semiconductor junction (0.45 V^−1^ to 1.5 V^−1^) [58]. For the high bias voltage (region III, 5.8 V < *V* < 9 V), the current transport characteristics of the n-Al:ZnO NRs/p-BDD heterojunction conform to the law *I~V*^1.93^, and this relationship is usually related to the space charge-limited current (SCLC) characteristics in wide bandgap semiconductors. When the exponent is almost equal to or greater than 2, the trap-limited SCLC conduction model coupled to the exponential distribution trap can explain it [59,60]. In general, structural defects located within the bandgap, the undoped ZnO and/or the diamond side generate more trap states to trap free carriers, which then make the concentration of injected carriers lower from diamond to ZnO at room temperature. When Al is doped into ZnO NRs the trap states will be filled by larger doped free carriers in the n-Al:ZnO NRs/p-BDD heterojunction. Therefore, in region III, the injection current follows an untrapped SCLC characteristic with an index close to 2.

## 3. Materials and Methods

Hot filament chemical vapor deposition (HFCVD) was used to make p-type BDD films with a thickness of ~4 μm on silicon substrates [61]. The synthesis of Al:ZnO NRs was performed using the hydrothermal method. Before developing Al:ZnO NRs, magnetron sputtering was used to create ZnO seed crystal layers on the diamond film with a thickness of 30 nm. Zinc acetate dihydrate, aluminum nitrate ninhydrates and anhydrous ethanol were combined in a 0.05 M solution and stirred with a magnetic stirrer until the components were completely dissolved. The precursor solution and the diamond film for developing the ZnO seed layer are transferred to the autoclave liner once the ingredients have been completely dissolved, and a little amount of sodium hydroxide solid is added as the mineralizing agent. The diamond films with Al:ZnO NRs were removed from the oven after 24 h of treatment at 150 °C, and the samples were repeatedly rinsed with anhydrous ethanol solution and dried at room temperature.

The morphology and structure of the samples were examined using scanning electron microscopy (SEM, Carl Zeiss, Oberkochen, Germany). The elemental composition of the samples was analyzed by energy-dispersive X-ray spectroscopy (EDS, Carl Zeiss, Oberkochen, Germany). The phase structure and phase purity of the samples were examined using X-ray diffractometry with Cu K_α_ radiation (XRD, Rigaku SmartLab, Tokyo, Japan). PL properties were characterized using FLS920 spectro-fluorophotometer (Edinburgh Instruments, Edinburgh, UK). The *I-V* characteristics of the heterojunction were measured using a Keithley 2400 source (Keithley Instrument, Cleveland, OH, USA).

## 4. Conclusions

In conclusion, we used a hydrothermal technique to construct an n-Al:ZnO NRs/p-BDD heterojunction and examined its structural, optical and electrical properties. The grown Al:ZnO NRs were well shaped and had a fibrillated zincite hexagonal phase. The PL studies show that the Al doping increases the oxygen hole concentration of the heterojunction, which indirectly affects the luminescence intensity and widens the band gap of ZnO, leading to a significant blue shift of the UV emission peak. The fabricated heterojunction devices exhibit excellent electrical properties such as lower turn-on voltage (0.27 V), smaller reverse leakage current (0.077 μA) at −5 V, large rectification ratio (838) and high forward current (67.5 mA) at +5 V. The barrier height of the heterojunction at room temperature decreases after Al doping is attributed to the structural defects and doping level. This work contributes to the field of diamond-based device design and application by providing a high-performance device for the development of future optoelectronic nanodevices.

## Figures and Tables

**Figure 1 ijms-23-03831-f001:**
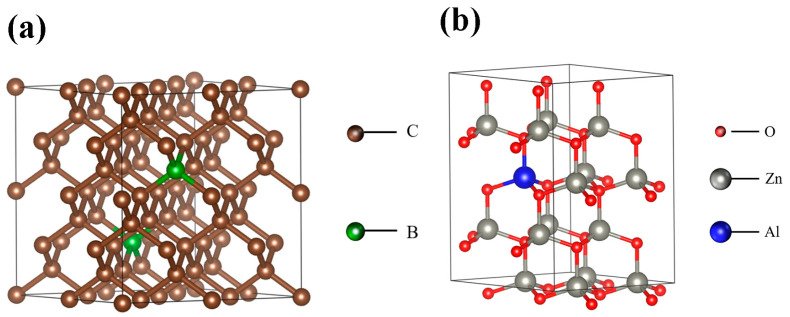
(**a**) Crystal structure diagram of BDD. (**b**) Crystal structure diagram of Al:ZnO NRs.

**Figure 2 ijms-23-03831-f002:**
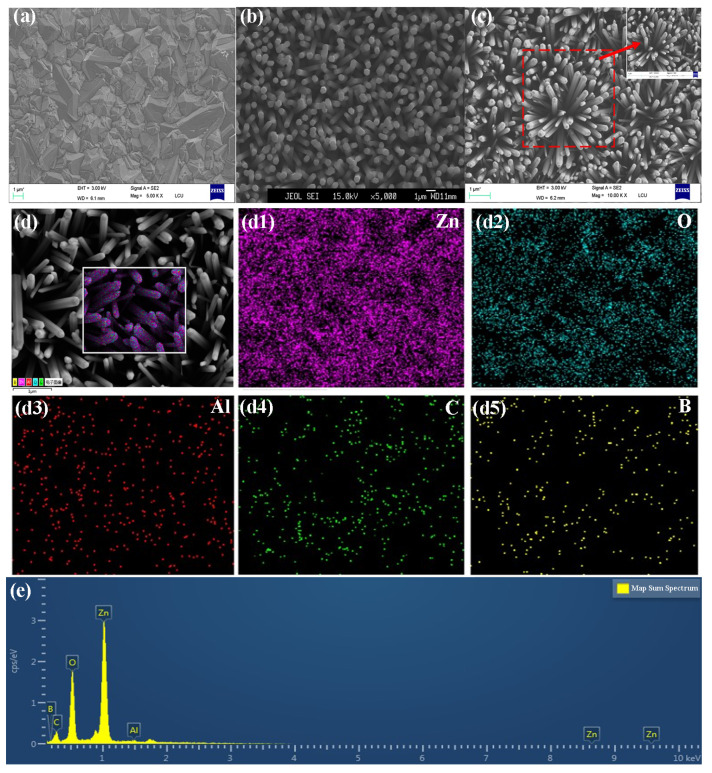
(**a**) SEM of a planar vertical view of the BDD substrate. (**b**) SEM of ZnO NRs deposited on the BDD substrate. (**c**) SEM of Al:ZnO NRs deposited on the BDD substrate, with the illustration being a magnified SEM image of the selected section. (**d**) EDS layered image of Al:ZnO NRs deposited on the BDD substrate. (**d1**–**d5**) the corresponding EDS mappings of (**d1**) Zn (**d2**) O (**d3**) Al (**d4**) C and (**d5**) B. (**e**) EDS spectra of n-Al:ZnO NRs/p-BDD heterojunction.

**Figure 3 ijms-23-03831-f003:**
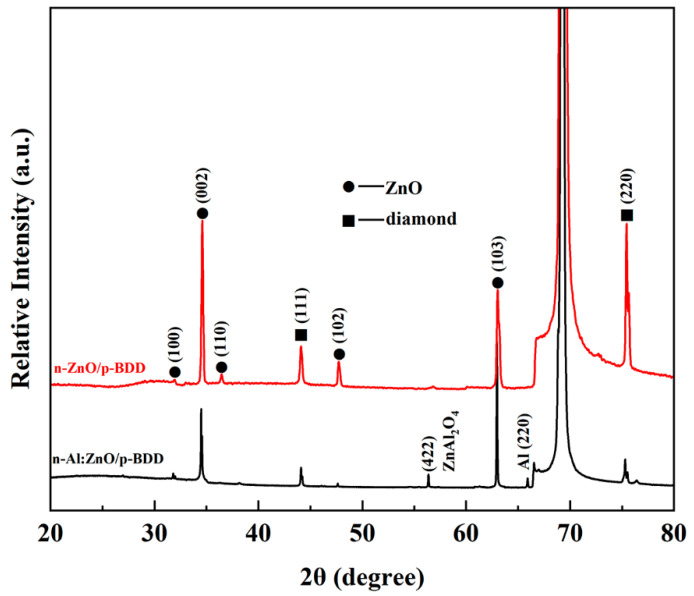
XRD pattern of n-ZnO NRs/p-BDD heterojunction and n-Al:ZnO NRs/p-BDD heterojunction.

**Figure 4 ijms-23-03831-f004:**
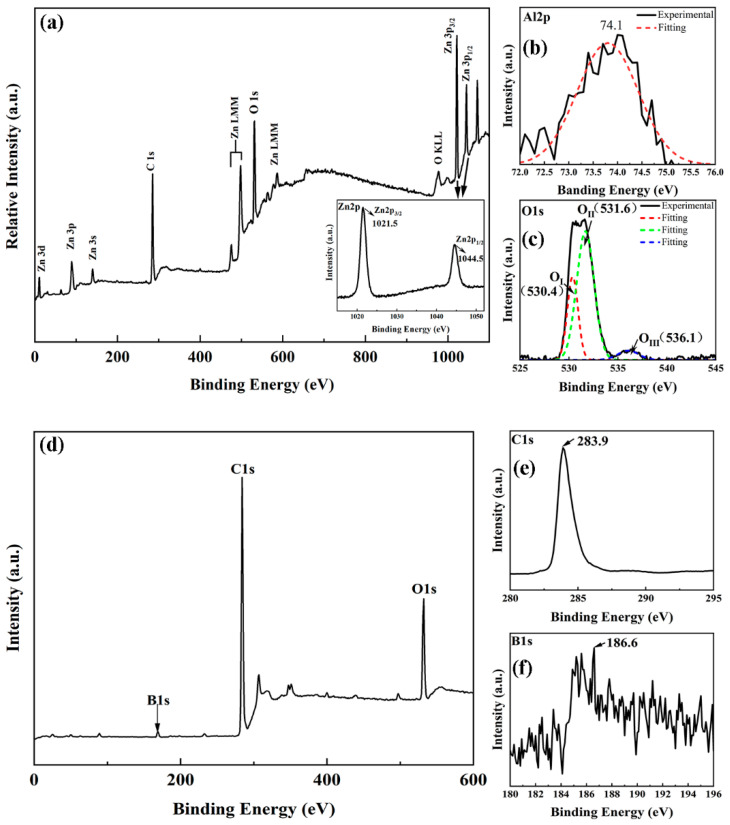
(**a**) XPS spectra of n-Al:ZnO/p-BDD heterojunction; the inset shows the high-resolution XPS spectrum of Zn2p. (**b**,**c**) High-resolution XPS spectra of Al2p and O1s, respectively. (**d**) XPS spectra of BDD substrate. (**e**,**f**) High-resolution XPS spectra of C1s and B1s, respectively.

**Figure 5 ijms-23-03831-f005:**
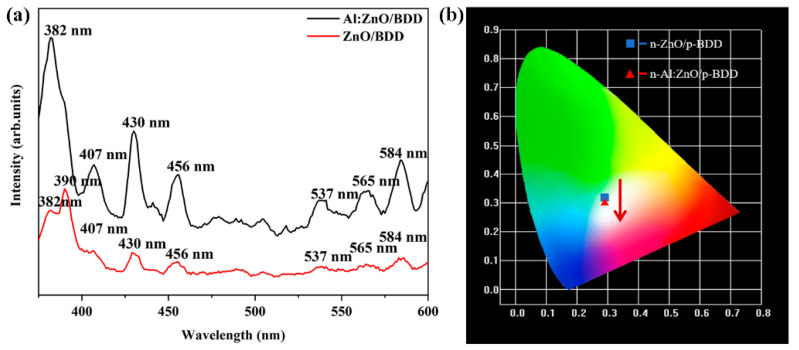
(**a**) Comparison of room temperature PL spectra of n-Al:ZnO NRs/p-BDD heterojunctions with n-ZnO NRs/p-BDD heterojunctions. (**b**) The CIE chromaticity maps were transformed from the left graph. A laser with a wavelength of 346 nm is used as the excitation source.

**Figure 6 ijms-23-03831-f006:**
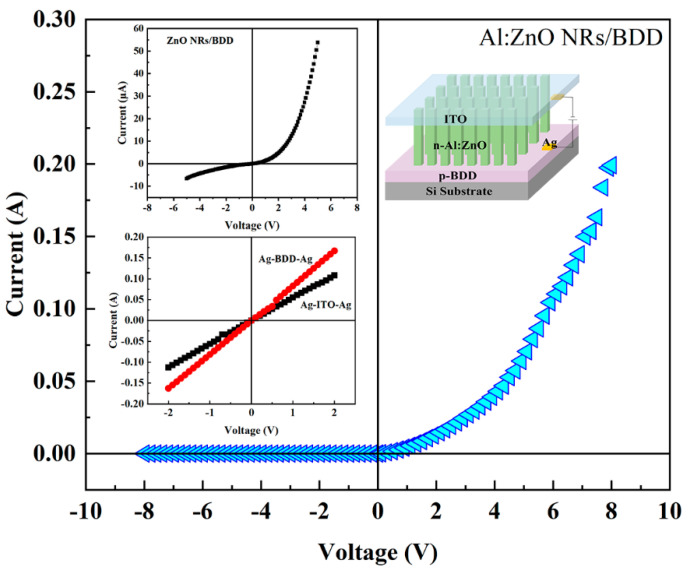
*I-V* characteristics of n-Al:ZnO NRs/p-BDD heterojunction at room temperature. The top left illustration shows *I-V* characteristics of n-ZnO NRs/p-BDD heterojunction at room temperature. The top right illustration shows the schematic diagram of the n-Al:ZnO NRs/p-BDD heterojunction device. The bottom left illustration shows the ohmic contact test for Ag/ITO/Ag and Ag/BDD/Ag.

**Figure 7 ijms-23-03831-f007:**
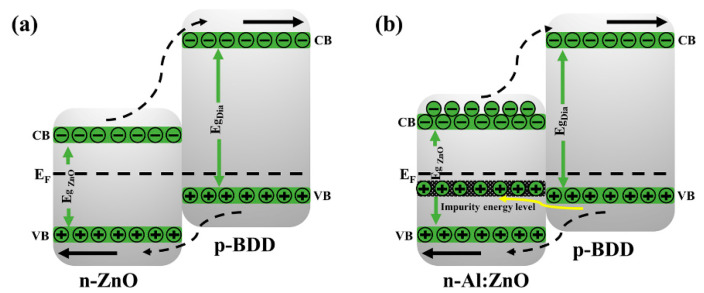
Energy band structure diagrams of (**a**) n-ZnO NRs/p-BDD heterojunction and (**b**) n-Al:ZnO NRs/p-BDD heterojunction.

**Figure 8 ijms-23-03831-f008:**
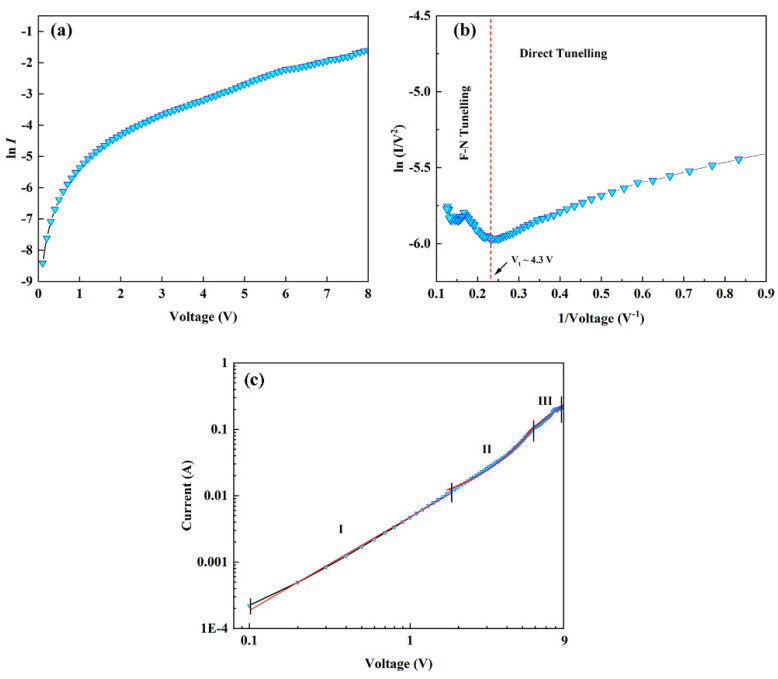
Plots of (**a**) ln*I-V* and (**b**) ln (*I/V*^2^ vs. 1/*V*) for Al:ZnO NRs NRs/BDD heterojunction. (**c**) log*I*–log*V* plots of n-Al:ZnO NRs/p-BDD heterojunction.

## Data Availability

The data presented in this study is contained within the article.

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
