# Peer review of "Enhanced Photoluminescence and Electrical Properties of n-Al-Doped ZnO Nanorods/p-B-Doped Diamond Heterojunction"

_ijms, 2022, doi:10.3390/ijms23073831_

Round 1

Reviewer 1 Report

Review of the manuscript-ijms-1635560-v1-to Authors: This paper studies optical and electrical properties of aluminium doped ZnO/boron doped diamond heterojunction. The paper offers sufficient novelty only in terms of combination of AZO with the boron doped diamond and the resulting n/p heterojunction.

In introduction please expand a bit on why boron doped diamond is better than other options for the heterojunction. Al doping: “tiny ionic radius” replace with “small ionic radius”.

In the results section the part where you write about the crystal structures and include the include them in Fig.1 is in my opinion better suited in the introduction, since this is not a direct result from your measurements.

“The Al:ZnO NRs maintain the hexag-114 onal fibrillated ZnO structure, which indicates that the doping of Al ions does not change 115 the crystal structure of ZnO. In addition, the (100), (002), and (103) diffraction peaks as-116 sociated with ZnO in Al:ZnO NRs are found to shift to lower angles, which may be due to 117 the substitution of Al3+ for Zn2+ in the crystals [27].” it doesn’t affect, but it does affect? Maybe expand or rewrite this sentence.

“Therefore, the variation of the inten-144 sity of this peak reflect the variation of oxygen vacancies within the heterojunction” Maybe add some reference/explanation for this.

“and may be related to 150 the variation of Al atoms within the different lattices” this sentence is also a bit ambiguous, please explain further.

“By introducing an 165 exotic dopant aluminum, the defect environment will change whether the aluminum 166 atoms replace the zinc atoms or occupy the gap positions” I would not really call Al an exotic dopant, especially for ZnO

“The heterojunction has good rectification characteristics with a high rectification ratio 214 of 838 at 5 V, which is more than 40 times higher than that of the n-ZnO NRs/ P-BDD 215 heterojunction (19.3) [22].” Why didn’t you prepare a sample without Al and measure comparatively, but choose to use literature values?

Having in mind the only interesting and novel thing in this paper is the actual heterojunction forming, I have t generally consider two issues; what shows that you indeed have a fully developed heterojunctions, and second, what did you actually want to show when selecting this system, namely there are many strategies available to reach the same, and some people might say they are more easily reachable.

Language throughout the manuscript could also benefit from an upgrade. In my opinion the paper will reach the level suitable for publishing, but after a several upgrades, so I recommend major revision.

Reviewer 2 Report

The manuscript has been well and systematically organized. Experimental results are interesting to other researchers in related fields. However, it is necessary to include some modification with the further scientific discussions. When you revise the manuscript, please check the below items:

  1. Page 2, line 74. The authors presented that the Al:ZnO NRs are oriented mostly vertically on the diamond substrate… (a) Such description does not correspond to the Fig. 1 (b). (b) I recommend that the plan-view SEM image of ZnO:NRs deposited on diamond substrate should be included in the Fig. 1 for improving the readability.
  2. Page 4. The intensity of XRD signal of the Al:ZnO NRs is weaker than that of the ZnO NRs. What are the reasons?
  3. The average crystalline size estimated from XRD data should be reported in this work.
  4. I-V curve of the n-Al:ZnO NRs/p-BDD heterojunction device should be included in the Fig. 5.
  5. Characters are too small in some figures.

Round 2

Reviewer 1 Report

Review of the manuscript-ijms-1635560-v2-to Authors: This is the revised version of the paper that studies optical and electrical properties of aluminium doped ZnO/boron doped diamond heterojunction. The authors answered all of the queries and improved the quality of the manuscript. The remaining issues can be dealt with in another proofreading. I think the manuscript is suitable for publishing.

Reviewer 2 Report

The authors answered appropriately the remarks. This version of the manuscript was accordingly revised and improved.